# ENHANCING LLM FACTUALITY FOR STRUCTURED DATA

## ABSTRACT

Large language models (LLMs) are typically optimized to process and output high-quality unstructured text, demonstrating remarkable capabilities in a variety of natural language tasks. Yet in practical settings, many domains, such as safety-critical or enterprise applications, rely on structured data. Improving the factuality of contemporary LLMs in these scenarios remains an open challenge, given their propensity to hallucinate or generate incorrect responses. In this work, we propose a methodology to enhance the factuality of LLMs when plugging into structured data. Specifically, we design a method for verbalizing proprietary structured data in a way that it is presented to LLM in longer context paragraphs, with a strong focus on the generation of sophisticated adversarial paragraphs that improve the LLM's resilience to hallucination and help detect factual errors in current solutions.

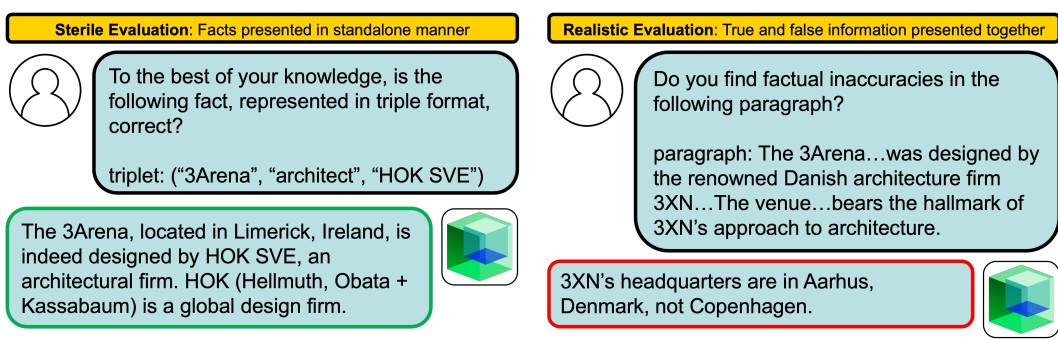

Figure 1: Despite understanding the factual truth under simple evaluation settings, the same large language model may fail to utilize this fact to accurately identify misinformation within context. Note that additionally, the language model's correction (right) is also incorrect - 3XN's headquarters are indeed in Copenhagen.

## 1 INTRODUCTION

The capabilities of large language models (LLMs) have progressed at a rapid pace, becoming integral components for a wide range of applications, tools, and general day-to-day usage (Team, 2024; AI, 2024; Anthropic, 2024; OpenAI, 2023; Li, 2025; Nam et al., 2024; Wu et al., 2024; Yuan et al., 2025). As LLMs are increasingly integrated into enterprise systems, their interaction with structured and proprietary data has emerged as a focal point. Organizations are exploring ways to securely leverage internal databases and confidential datasets to enhance model relevance, while maintaining strict data governance and guaranteeing the correctness and factuality of model outputs – particularly when operating in high-stakes domains such as finance, healthcare, or law, to name a few.

A prominent challenges when relying on LLMs to integrate and deliver proprietary data to consumers is that LLMs face significant difficulties in accurately discerning misinformation, even when accurate data has been included in their training procedure. This issue is not merely a limitation of coverage but stems from the statistical nature of language modeling itself. Since LLMs learn

from patterns in data rather than understanding facts in a human-like way, they may fail to distinguish between high-probability sequences that are accurate and those that are subtly misleading or false (Chen & Shu, 2024). This challenge becomes even more pronounced when misinformation is embedded within longer or complex textual contexts. In such cases, the truth signal may be diluted, allowing falsehoods to blend in and pass as plausible, especially if those falsehoods co-occur with factual statements. Another exacerbating factor is the nature of the data we want to guarantee correctness about. While dealing with common and open source knowledge, assessing its correctness can exploit statistical properties within the data. On the other hand, enterprise data, which is often proprietary, requires a systematic way to test and assess the factuality of a model before deploying it for specific business applications is paramount. Moreover, LLMs are typically more effective at recognizing and reaffirming known correct information than at identifying and rejecting incorrect or misleading statements. This asymmetry arises because confirming known facts often involves retrieving memorized patterns or repeated associations, while detecting plausible misinformation requires more nuanced reasoning and the ability to detect inconsistencies or anomalies (Guo et al., 2022). As a result, even when correct data is present in the training set, its statistical weight may be insufficient to counteract or displace more frequently occurring but incorrect associations. This fundamental vulnerability underscores the need for more robust methods of grounding and fact-checking LLM systems. All these challenges make it extremely difficult to assess and guarantee the correctness of the generated data, especially when the target data is proprietary or uncommon, which is not covered by standard factuality benchmarks for LLMs.

Our solution addresses the challenge of factual consistency in LLM outputs by leveraging proprietary structured data known to have been a part of the training distribution. We introduce a systematic methodology that allows the model to self-verify its responses against this structured data, enabling a form of internal correctness assurance without relying on external evaluators. Our method assumes access to a set of proprietary structured data, i.e., database tables, knowledge graph triples, or in general a collection of structured/semi-structured facts. From the proprietary knowledge base, we systematically generate a series of subtly incorrect and/or misleading assertions about the data, relying on Typed Constrained Negative Sampling (TCNS) (Krompaß et al., 2015; Han et al., 2024; Qiu et al., 2024; Bai et al., 2022; Ahrabian et al., 2020; Yang et al., 2024). We use both original (correct) and the perturbed assertions to generate coherent paragraphs, which are then used to (i) fine-tune the model and to (ii) systematically assess the model's capacity to distinguish between accurate and subtly misleading content. Our pipeline can synthetically generate high-quality training data derived from any given proprietary structured data source; this data is designed to reflect the structure, semantics, and critical facts embedded in the data source, ensuring alignment with ground truth. By fine-tuning the model on this curated dataset, we significantly enhance its robustness to misinformation, improving both factual accuracy and resistance to hallucination.

This work makes several contributions. First, we aim to bridge the gap between symbolic data representation and neural language models: we introduce a method to systematically transform structured data into a format that can be effectively processed by LLMs; our novel methodology embeds structured facts into cohesive paragraphs that are conducive to enhancing model robustness. Second, we propose a methodology for improving model robustness through the dynamic generation of challenging benchmarks, leveraging structured data such as knowledge graphs to produce plausibly in-domain false triples that are close enough to the correct facts to challenge the model behaviors. Third, we present a fine-tuning strategy designed to enhance factual accuracy, yielding improved performance in factual consistency tasks.

Our approach is particularly well-suited for poorly covered domains, especially where available data is legacy and private. Its plug-and-play nature makes it highly adaptable, requiring minimal changes to existing pipelines, with the only constraint being the presence of structured data as input. A key advantage of our method lies in its dynamic synthetic data generation, which produces text in a format and style tailored specifically to feed and extend the underlying LLM. This is enabled by the novel verbalization strategy that intelligently selects the most effective way to express each paragraph, thereby maximizing model robustness and performance, even in previously unseen scenarios.

## 2 RELATED WORK

### 2.1 FACTUALITY IN LLMs

Factuality in LLMs refers to their ability to produce accurate and reliable outputs. Many methodologies and metrics focus on assessing and quantifying the amount of LLM-produced output that is inconsistent with established facts (Wang et al., 2025; Augenstein et al., 2024), using tailored benchmark datasets for misinformation and hallucination detection (Bang et al., 2025; Friel & Sanyal, 2023; chen et al., 2023) and large-scale evaluations (Fu et al., 2023; Wang et al., 2024; He et al., 2024b). Available frameworks/systems (Iqbal et al., 2024; Marinescu et al., 2025) and solutions (Cohen et al., 2025; Lin et al., 2024) aim at LLM factuality alignment. Some methods rely on structured data in the form of KG, either as an external source of factual triples (Xu et al., 2024) or as a source of factual context for LLM prompts (Perozzi et al., 2024). Novel solutions such as Factoscope (He et al., 2024a) leverage the inner states of LLMs for factual detection, as they demonstrate that models exhibit distinguishable patterns in their inner states when generating factual versus non-factual content. While this is a promising approach, it requires access to the model's inner states and is not model-agnostic. The majority of works on factuality focus on the assessment of single facts, and even benchmarks for long-form factuality, such as FactoScope (Wei et al., 2024), break down the long-form response of an LLM into a set of individual facts and evaluate each fact separately. Our approach is interested in methodologies for assessing and improving factuality, in settings where incorrect facts are (i) proprietary, previously unseen by the model structured data, that are (ii) naturally embedded in longer text paragraphs.

### 2.2 LLMs AND STRUCTURED DATA

LLM's lack of factuality is a clearly and publicly perceived issue in open and general domains. But the problem is exacerbated in business settings when the new wish and trend is to access, interface, and serve proprietary data via LLM's interactions. LLMs have known limitations in truly comprehending structured data, e.g., interpreting tables (Sui et al., 2024) or reasoning over graph data (Guo et al., 2023). Many recent efforts aim at bridging this gap by incorporating structured knowledge to enhance model performance and development. Knowledge graphs have been utilized for factuality assessment (Liu et al., 2024; Luo et al., 2023; Feng et al., 2023), verification (Opsahl, 2024; Vedula & Parthasarathy, 2021; Ribeiro et al., 2022), and dataset creation (Tchechmedjiev et al., 2019; Song et al., 2023), among many more. Another notable example of interactions of LLM with structured data is StructGPT(Jiang et al., 2023) that collects relevant evidence from structured data and lets LLMs concentrate on the reasoning task based on the collected information.

Our method utilizes structured data - in the form of KG - in a novel way, i.e., embedding all the structured triples in longer paragraphs to fine-tune the models. A similar idea was proposed by (Patel et al., 2025) that uses semantic operators to transform structured data using natural language specifications (e.g., filtering, sorting, joining, or aggregating records using natural language criteria). Our major difference with the state of the art is the generation of both correct paragraphs, but also paragraphs containing wrong facts according to the source data, which are very difficult to identify. Unlike prior works, knowledge graphs are only a tool that we use to generate candidate misleading assertions, which are then used as seed exemplars to generate our longer contexts. Embedding these assertions within longer textual contexts results in more realistic and difficult samples for factuality assessment.

### 2.3 NEGATIVE SAMPLING

Negative sampling is a widespread technique with applications across various fields, including machine learning, inductive logic programming (Sen et al., 2020; Sadeghian et al., 2019), computer vision, natural language processing, data mining, recommender systems, just to mention a few. When it comes to negative sampling over structured data, especially knowledge graphs, the advantage comes from the availability of out-of-the-box typing constraints in the data (Krompaß et al., 2015). Several standard techniques have been explored in literature, including: (i) naively perturbing either the subject or the object with a random entity from the entire KG (Zhang et al., 2019); (ii) contrastive learning, i.e. selecting negative samples that are more difficult to discriminate using either some arbitrary properties of the relations within the graph (Ebisu & Ichise, 2018) or exploit-

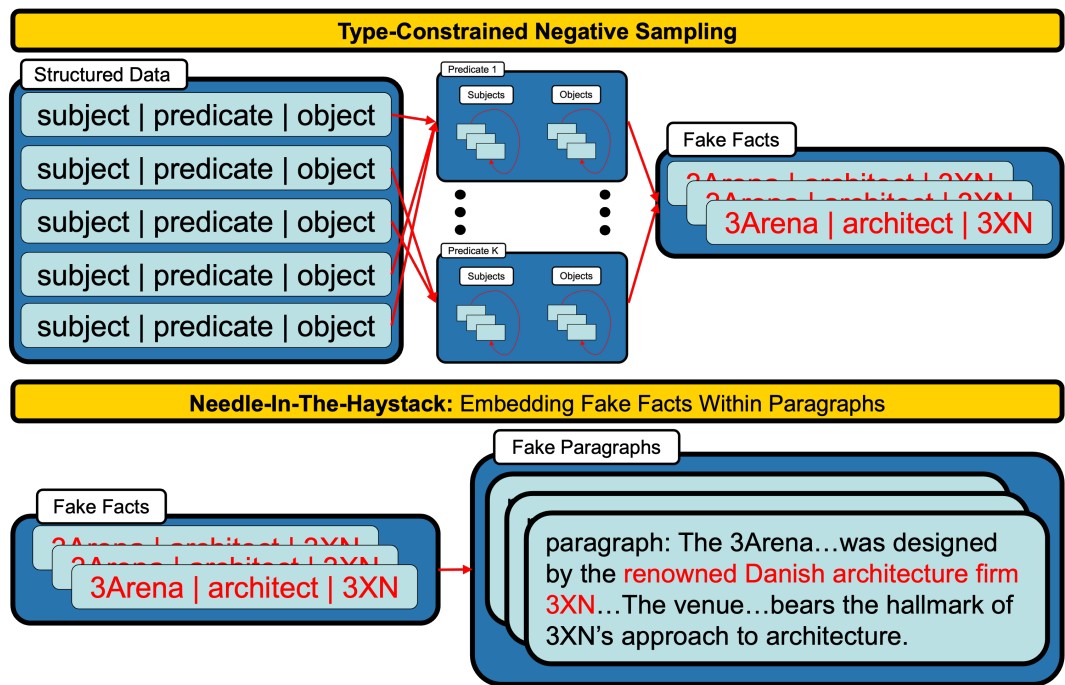

Figure 2: (Top) Using an existing knowledge base, we generate type-constrained negative samples to create plausible but fake facts. (Bottom) Fake facts are used as the seed to embed misinformation into longer paragraph contexts.

ing the the graph topology(Ahrabian et al., 2020); (iii) using type constraints to reduce the scope of the negative sample (Krompaß et al., 2015); (iv) learning adversarial models to generate the negative examples (Cai & Wang, 2018) or (v) exploiting the proximity of entities in their embedding space (Islam et al., 2022). Further sophistication includes pre-filtering easy-to-discriminate negatives (Han et al., 2024) to reduce the number of candidates the model needs to compute during training - the focus in this case is on minimizing computation. Others have proposed an approach designed to synthetically generate open category samples without requiring any prior knowledge or external datasets (Bai et al., 2022). All these mentioned approaches work on a one triple/statement at a time, hence they are very effective for tasks like link prediction (Hubert et al., 2022), but not so suited for general LLMs, where facts, statements, and assertions are always embedded in a larger text context.

The most similar related work also focuses on the perturbation of paragraphs (Qiu et al., 2024). They generate perturbed summaries with a multi-step process: they start from a paragraph, convert it into a graph, inject controlled factual inconsistencies in the graph, and then create negative examples. The multi-step nature of this work is prone to errors, and any mistake made in the graph extraction would be propagated in the subsequent steps, making it hard to systematically track. In contrast, our work combines the systematic nature of the typical typed constrained negative sampling approach, but embeds the fake facts in sophisticated long paragraphs, hence generating robust synthetic data for LLM settings. To the best of our knowledge, we are the first to propose a method combining these two aspects.

## 3 LEVERAGING KNOWLEDGE GRAPHS FOR FACTUALITY ENHANCEMENT

We leverage the presence of an available knowledge base to enhance the factuality performance of language models (LMs). First, we utilize structured data (e.g. knowledge graphs) to generate type-constrained negative samples, which are our fake facts. Second, we embed the generated facts into longer paragraph contexts. Finally, we design a task around these paragraphs and finetune LMs

on this data, demonstrating enhanced factuality performance. An illustration of our data generation procedure is provided in Figure 2, and we describe each component in the following sections.

### 3.1 GENERATING PERTURBED FACTS

We leverage an existing knowledge base to generate our fake facts. Our objective is to create fake facts that are hard negative samples for the downstream LM, yet are plausible and sound potentially truthful. Given an input knowledge base, we first group all samples in our dataset into various buckets. These buckets are categorized and distinguished by the knowledge graph predicates, or edge relations, in our dataset. Each bucket corresponds to exactly one predicate, and the number of buckets is equal to the number of unique edge relations that exist in our dataset.

In each bucket, we then randomize the list of subjects and objects within each other - this is done by shuffling all subjects and objects within themselves. While we may shuffle subjects with other subjects, and likewise for objects, note that we *do not* shuffle subjects and objects between each other. Finally, we then filter our any generated facts which can be found in the original dataset. This is because since we are shuffling lists of subjects and objects (and not sets), there is the slim possibility that we retain an original truthful fact in our list of generated fake facts.

Our procedure to generate fake facts is done entirely without the use of any LLMs or foundation models, eliminating the possibility of any errors arising due to model hallucinations. Of course, our procedure is based on the assumption that the original dataset contains true facts only with a few errors. We believe that for many domains and applications, particularly safety-critical and enterprise settings, this is a reasonable assumption to make of the datasets.

### 3.2 THE NEEDLE IN A HAYSTACK

One of the core novelties in our method lies in how we embed our fake facts into longer paragraph contexts. Certainly, this task can be performed in a zero-shot manner with simple LM prompting. However, this (i) does not guarantee that the output paragraphs will still contain the seed fake fact that was used, and (ii) might create trivial paragraphs, where the task of identifying the perturbed facts is very easy for the model. Our aim is to generate paragraphs that clearly contain one fake fact and do an adequate job at camouflaging such a fact, making the identification task difficult for LMs that do not contain the requisite knowledge. For this purpose, we design a procedure to tune an LM as a fake-fact paragraph generator using GRPO (DeepSeek-AI et al., 2025a; Shao et al., 2024) - our goal is to generate a paragraph containing a fake fact, which camouflages the fact in a plausible way. In this manner, models that are not equipped with the fact itself, for example, an off-the-shelf LM, would not be able to accurately identify the fake fact.

We denote our paragraph generator as $\text{LM}_{\text{gen}}$ and assume this to be initialized from an off-the-shelf LM. We initially generate a paragraph $p$, containing a fake fact, with simple prompting to $\text{LM}_{\text{static}}$. We then use another static instance of the same off-the-shelf LM, $\text{LM}_{\text{static}}$, and we prompt it in two different ways: student(p) is a prompt that simply presents the paragraph and asks for fake fact identification, while oracle(p) is a prompt that also tells the model the source fact before asking for fake fact identification within the paragraph.

The main function of oracle(p) is basically ensuring that the fake fact is indeed present in the generated paragraph $p$. This is important to ensure that the LM does not camouflage the fake fact by simply generating an irrelevant paragraph - a trivial yet erroneous solution. The desired characteristic of $p$ is that it can successfully fool $LM_{static}$ prompted with student(p), but not with oracle(p).

We formulate our reward function for our $\text{LM}_{\text{gen}}$ paragraph generator as follows:

$$\text{REWARD}(p) = \begin{cases} 1 & \texttt{LM}_{\texttt{static}}(\texttt{student(p)}) = 0, \texttt{LM}_{\texttt{static}}(\texttt{oracle(p)}) = 1 \\ 0 & \texttt{else} \end{cases}$$

Where $\texttt{LM}_{\texttt{static}}(\texttt{student(p)}) = 0$ indicates that the model - interrogated with a student prompt - was unable to identify the incorrect fact, and $\texttt{LM}_{\texttt{static}}(\texttt{oracle(p)}) = 1$ indicates that the model - interrogated with an oracle prompt - was able to correctly identify the incorrect fact. This reward helps incentivize paragraphs where it is difficult to identify the fake fact, without the paragraph

generator model diverging and generating paragraphs that no longer contain the original source fake fact.

Note that, unlike prior work, by generating the paragraphs only using a single fake fact as a seed, we eliminate the potential of contradictions arising in the generated paragraphs, which can happen when the method starts from a given paragraph and perpetuates some assertions in an unbounded way (i.e., without type constraints).

### 3.3 INJECTING STRUCTURED KNOWLEDGE

The generated data can be effectively used as a dataset and use-case-specific benchmarking tool, obviously using a determined split of data that remains unseen to the model. Outside of benchmarking and evaluation, we demonstrate that the generated paragraphs are an extremely effective way to pass structured knowledge to LLMs for supervised fine-tuning (SFT), as passing data in context is consonant with the way LLMs learn. We model the factuality alignment procedure as a text generation task, where the model is optimized to generate the source fact that was used to seed the generation of the sample paragraph. One example of our task is seen from the failure mode of the LM in Figure 1.

Unlike baseline approaches, our generated data serves to augment the original knowledge base and create multi-view training data. Training via negative samples has long been used in literature and has been proven to improve model performance for key foundational tasks, such as word2vec (Mikolov et al., 2013). By grouping samples within relation, we are able to ensure that our negative samples are more plausible and harder to distinguish from truthful facts than simple prompt-based approaches.

## 4 EXPERIMENTAL SETTINGS AND RESULTS

We evaluate the performance of various LMs on structured knowledge factuality. Please refer to the Appendix for model descriptions, prompts, hyperparameters, or other experimental parameters.

### 4.1 DATASETS

We employ two RDF[1] triple datasets: WebNLG and Rebel, which contain knowledge graphs about various facts (Gardent et al., 2017; Huguet Cabot & Navigli, 2021). For the WebNLG dataset, we use the English training split as our structured knowledge base, which contains a total of 13211 triples (3501 unique) and 360 unique relations. For the Rebel dataset, we use the training split as our structured knowledge base, and we randomly select 3000 samples containing a total of 7915 triples (7369 unique), and 268 unique relations.

From the WebNLG dataset, we generate a total of 3407 triples corresponding to fake facts, while for the Rebel dataset we generate a total of 6957 triples corresponding to fake facts. We evaluate our experiments considering Granite-3.3-8B-Instruct and also Llama-3.1-8B-Instruct as our static paragraph generator models ($LM_{static}$).

### 4.2 RESULTS AND DISCUSSION

**[Topline] Factuality Assessment**  Rather than a *baseline*, in our work we use a *topline* for comparisons. We refer to this setting as *topline*, because, while describing the most basic and easiest evaluation setting, it is the one that theoretically should result in the highest achievable performance: the easier the questions, the higher the likelihood of the LM to get them right.

Our *topline* consists of three straightforward steps. First, we take the original facts expressed in triple form. Next, we construct a simple zero-shot prompt which merely presents the fact and asks "Is this fact correct?" without any additional context or fine-tuning (see the left portion of Figure 1. Finally, we submit this prompt as is to the LLM and verify whether the LLM is capable of accurately assessing the veracity of the given facts. This setup allows us to evaluate the model's reasoning ability in its most direct and unassisted form.

---

[1] https://www.w3.org/RDF/

| Model | Topline | Zero-Shot Generated Paragraphs | | GRPO Generated Paragraphs | |
|---|---|---|---|---|---|
| | Simple Prompt | Granite-8B | Llama-8B | Granite-8B | Llama-8B |
| Granite-8B | 67.10% | 78.16% | 77.31% | 71.24% | 73.44% |
| Llama-8B | 62.67% | 65.66% | 66.13% | 62.81% | 69.18% |
| Qwen3-8B | 71.95% | 64.25% | 77.84% | 68.56% | 76.99% |
| Phi-4 | 62.18% | 83.21% | 82.80% | 80.33% | 81.51% |
| Qwen-72B | 49.79% | 83.56% | 80.92% | 76.70% | 82.74% |

Table 1: LLM fact identification performance on the WebNLG source dataset. The left-most column contains results for correct fact identification (*topline*). The other sections denote results on fake fact identification for zero-shot generated paragraphs (generating paragraphs via prompting LLMs) and GRPO-generated paragraphs (generated paragraphs from our GRPO-tuned model).

| Model | Topline | Zero-Shot Generated Paragraphs | | GRPO Generated Paragraphs | |
|---|---|---|---|---|---|
| | Simple Prompt | Granite-8B | Llama-8B | Granite-8B | Llama-8B |
| Granite-8B | 76.70% | 77.02% | 69.89% | 63.30% | 72.86% |
| Llama-8B | 34.25% | 64.89% | 64.07% | 57.38% | 70.13% |
| Qwen3-8B | 66.86% | 69.95% | 66.46% | 56.18% | 70.48% |
| Phi-4 | 66.16% | 85.27% | 80.72% | 79.81% | 84.66% |
| Qwen-72B | 47.40% | 83.97% | 81.14% | 77.38% | 86.32% |

Table 2: LLM fact identification performance on the Rebel source dataset. The left-most column contains results for correct fact identification (*topline*). The other sections denote results on fake fact identification for zero-shot generated paragraphs (generating paragraphs via prompting LLMs) and GRPO-generated paragraphs (generated paragraphs from our GRPO-tuned model).

As observed in Table 1, all tested models perform equivalently on the WebNLG dataset, with an average accuracy of 62.74% (with Qwen3-8B performing the best and Qwen-2.5-72B-Instruct performing the worst). For the Rebel dataset, performance is slightly more diverse, as evidenced by Table 2. In this case, Llama-3.1-8B-Instruct exhibits a notable dip in performance compared to the other models. During testing, we attributed this scenario to a peculiar case: the model will deny that a supplied fact is correct before providing a correction of the supplied information, which is exactly the same as the original input fact.

**[In-Paragraph] Factuality Assessment in a Plausible Paragraph**    In this setting, we investigate to what extent the model can identify a fake fact if it is embedded in a plausible paragraph, and we scrutinize whether the method used to generate the paragraphs impacts the results. Specifically, we compare the performance using paragraphs generated with our GRPO-based method and paragraphs generated with a zero-shot method. Our ultimate question is whether the paragraphs are sophisticated enough (i.e., GRPO-generated), would the LLM be fooled and miss the presence of a fake fact? On an overall glance, our experiments show that indeed all the tested LLMs tend to have more difficulty identifying fake facts within GRPO-generated paragraphs, as opposed to simple zero-shot generated paragraphs.

Table 1 and Table 2 show the degree of degradation in performance accuracy when tested on the Web-NLG and the REBEL datasets, respectively, with similar trends on both. Granite-3.3-8B-Instruct and Phi-4 drop by an average of 5.40% and 2.09%, respectively, compared with zero-shot generated paragraphs. Llama-3.1-8B-Instruct performs worse on Granite-generated paragraphs, dropping by 2.85%, while Qwen-2.5-72B-Instruct drops by 6.86%. On the other hand, Qwen3-8B performs relatively similarly across both scenarios. One interesting observation, however, is that on the Rebel dataset, all models perform worse on our GRPO-generated paragraphs, when the generator model uses Granite-3.3-8B-Instruct, with an average performance drop of 9.41%. Conversely, however, all models actually perform *better* when the generator model uses Llama-3.1-8B-Instruct, with an average performance gain of 4.43%. The usage of different paragraph generator modules may result in varying levels of difficulty in the generated paragraphs.

| Model | Topline | Fake Fact Identification Accuracy | |
|---|---|---|---|
| | Simple Prompt | Granite-8B | Llama-8B |
| SFT on Zero-Shot (Granite-8B) | 91.30% | 87.83% | 87.47% |
| SFT on Zero-Shot (Llama-8B) | 90.30% | 86.95% | 88.99% |
| SFT on Zero-Shot (Qwen3-8B) | 91.40% | 98.50% | 87.88% |
| SFT on GRPO (Granite-8B) | 93.80% | 94.86% | 87.03% |
| SFT on GRPO (Llama-8B) | 88.52% | 99.18% | 89.26% |
| SFT on GRPO (Qwen3-8B) | 79.61% | 91.87% | 86.41% |

Table 3: LLM SFT fact identification performance on the WebNLG source dataset. The left-most column contains results for correct fact identification (*topline*). The other columns contain results for fake fact identification, with the column denoting the source model that was used to generate the paragraphs ($LM_{static}$). The top split shows results when we fine-tuned LLMs on zero-shot generated paragraphs. The bottom split shows results when we fine-tuned LLMs on GRPO-generated paragraphs.

**[Fine-Tuned-Models] SFT Using GRPO Paragraphs**    Outside of dynamic factuality evaluation, we also measure the efficacy of our GRPO-generated paragraphs by observing the benefits they bring to models during SFT. Our GRPO-genereted paragraphs generally improve the overall performance when compared to SFT on zero-shot paragraphs, on both WebNLG (Table 3) and REBEL (Table 4) datasets.

Specifically, on WebNLG (Table 3) we see improvements for Granite-3.3-8B-Instruct in both correct fact recognition as well as misinformation detection on Granite-3.3-8B-Instruct generated paragraphs, improving by 2.50% and 7.03%, respectively. We also see a massive improvement when evaluating Llama-3.1-8B-Instruct on Granite source paragraphs, with performance improving by 12.23%. Interestingly, we see that Qwen3-8B degrades compared to baseline SFT, but this quirk may be due to stochasticity in the training process interfering with its inherent reasoning procedure.

| Model | Topline | Fake Fact Identification | |
|---|---|---|---|
| | Simple Prompt | Granite-8B | Llama-8B |
| SFT on Zero-Shot (Granite-8B) | 91.46% | 90.97% | 80.43% |
| SFT on Zero-Shot (Llama-8B) | 83.70% | 91.18% | 82.57% |
| SFT on Zero-Shot (Qwen3-8B) | 92.06% | 84.45% | 79.12% |
| SFT on GRPO (Granite-8B) | 93.84% | 93.34% | 81.64% |
| SFT on GRPO (Llama-8B) | 13.27% | 93.65% | 81.82% |
| SFT on GRPO (Qwen3-8B) | 86.12% | 86.58% | 75.79% |

Table 4: LLM SFT fact identification performance on the REBEL source dataset. The left-most column contains results for correct fact identification (*topline*). The other columns contain results for fake fact identification, with the column denoting the source model used to generate the paragraphs ($LM_{static}$). The top split shows results when we fine-tuned LLMs on zero-shot generated paragraphs. The bottom split shows results when we fine-tuned LLMs on GRPO-generated paragraphs.

On the REBEL dataset (Table 4), we observe the same trends, with Granite-3.3-8B-Instruct improving across the board, while Llama-3.1-8B-Instruct is able to improve by 2.47% on Granite-generated source paragraphs, as well as Qwen3-8B. We note that the poor correct fact performance of Llama-3.1-8B-Instruct is attributed to a peculiar behavior where it will state that the correct fact is wrong, but then it will state that it is correct. We evaluated model performance using a standard 80/20 split of the generated paragraphs as our train and test datasets, respectively.

## 5 CONCLUSION

In our work, we presented a methodology to dynamically generate negative samples from an input structured knowledge base. The data generation procedure, which does not require LLMs or any foundation models, generates negative samples via fake facts, exploiting the structure of the data

to group samples and generate plausible falsehoods. Additionally, we also formulated a novel task paradigm and injection scheme, using fake facts as seeds to embed misinformation into longer text contexts. We show the challenges of LLMs when it comes to identifying factual information in longer paragraph contexts, and also demonstrate the utility and effectiveness of our GRPO paragraph generation schema.

Several promising directions emerge from our study. We foresee exploring whether the same methodology can generalize effectively across diverse knowledge sources, in particular, temporal or multi-modal knowledge bases, which may yield new challenges and opportunities. We also plan on extending the evaluation to multilingual settings and low-resource languages. Another, more practical future work will involve positioning this technology within a comprehensive model guardrail pipeline, as well as the deployment for realistic applications, such as enterprise fact-checking and general enterprise information retrieval tasks. Studying how misinformation propagates within multi-turn conversations or across document networks may further illuminate the challenges of trustworthy AI in real-world deployments.

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

## A  MODELS

In our evaluations, we tested a total of 5 models, with 1 model as our LLM judge. These models are listed below:

**Granite-8B** denotes the Granite-3.3-8B-Instruct model, an 8B parameter dense decoder language model (Team, 2024).

**Llama-8B** denotes the Llama-3.1-8B-Instruct model, an 8B parameter dense decoder language model (AI, 2024).

**Qwen3-8B** is an 8.2B parameter dense decoder language model (Team, 2025).

**Prompts**

(Topline Prompt) To the best of your knowledge, is the following fact, represented in triple format, correct? Please ONLY answer 'yes' or 'no'.

triplet: $\{\mathcal{T}\}$

(LLM Judge Prompt) Given an associated fact, please answer if that fact is reasonably mentioned in the answer. Please ignore whether the fact is actually correct or not; only answer if it can be reasonably found in the provided answer.

associated fact: $\{\mathcal{T}\}$

answer: $\{A\}$

Please output only 'yes' or 'no' in your answer. Additionally, if the provided answer states that the given fact is correct, also answer 'no'.

(Zero-Shot Paragraph Generation Prompt) You are given a triple of the form [subject, relation, object]. Generate a paragraph that contains the information within the triplet.

triplet: $\{\mathcal{T}\}$

Your response should use the following format and include only the paragraph. The paragraph should be written in a factual tone. Do not mention the presence of the original triplet in your response.

paragraph: \<paragraph\>

(GRPO Oracle Template) Given the following paragraph, as well as an associated fact, please answer if that fact can be found in the paragraph.

$\{\mathcal{P}\}$

triplet: $\{\mathcal{T}\}$

Please output ONLY 'yes' or 'no' in your answer.

Table 5: All prompts that are used and/or relevant for our work. Note that $\mathcal{T}$ denotes the input triplet, $\mathcal{A}$ denotes an input answer (for the judge prompt), and $\mathcal{P}$ denotes an input paragraph (for the oracle prompt).

**Phi-4** is a 14B parameter dense decoder language model (Abdin et al., 2024).

**Qwen-72B** denotes the Qwen-2.5-72B-Instruct model, a 72B parameter dense decoder language model (Qwen et al., 2025).

**Deepseek-V3** is a mixture-of-experts transformer containing 671B total parameters, with only 37B parameters activated during inference per token (DeepSeek-AI et al., 2025b).

Note that for evaluating answer correctness, we employ LLM-as-a-judge for scalability and simplicity (Chiang & Lee, 2023). We use Deepseek-V3 as our judge model.

## B  MODEL PROMPTS

We detail all relevant prompts to our evaluation and data generation methodology in this section. Please refer to Table 5 for the collection of any relevant model prompts.

## C  EXPERIMENTAL SETTINGS

In this section, we detail any additional experimental settings that may provide benefit. For paragraph generation, we use sampling, with a maximum new tokens at 1024 and a minimum new tokens at 10, with a temperature of 0.7. For GRPO, we use a default of 1 epoch, with a learning rate of 2e-5, weight decay of 0.01, warmup ratio of 0.01, temperature of 0.7, top_p of 0.8, top_k of 50, with 4 iterations and 4 generations. Furthermore, we set the maximum prompt length to 256 and the maximum completion length to 512. For GRPO, we use a batch size of 2, with our gradient accumulation set at 8 steps, and a beta parameter of 0.04. For SFT, we use a default of 5 training epochs, with a learning rate of 2e-5, weight decay of 0.01, warmup ratio of 0.01, a maximum sequence length of 2048, and a batch size of 4 with 8 gradient accumulation steps. We use 42 for the value of any random seed.

