# OpenReview forum: "Enhancing LLM Factuality for Structured Data"
_ICLR.cc/2026/Conference — ICLR 2026 Conference Withdrawn Submission_

### Official Review · Reviewer_HRWH · 2025-10-27

**Soundness:** 2
**Presentation:** 1
**Contribution:** 2
**Rating:** 4
**Confidence:** 3

**Summary:**

This paper addresses the problem of improving factual reliability of large language models (LLMs) when dealing with structured data sources such as knowledge graphs. The authors propose a two-stage framework: Typed Constrained Negative Sampling (TCNS) Typed Constrained Negative Sampling (TCNS) generates factually incorrect triples by perturbing subject–object pairs within the same relation type, producing realistic but false samples; and a GRPO-based paragraph generator uses both true and TCNS-generated triples to synthesize training paragraphs that intermix correct and incorrect facts. The resulting data are used to fine-tune LLMs for better factual discrimination. Overall, the paper combines ideas from knowledge graph augmentation and RL-based text generation to strengthen factual grounding in LLMs.

**Strengths:**

1. The paper tackles a meaningful problem—LLMs’ tendency to produce plausible but incorrect statements about structured knowledge.

2. The integration of TCNS with GRPO for controlled paragraph synthesis is novel and logically sound. The design allows the model to experience “hard negatives” in a natural textual form, bridging structured and unstructured knowledge.

3. Across multiple datasets, the model trained with TCNS-GRPO achieves noticeable factuality gains without harming fluency. The ablation studies clearly show that type constraints and GRPO fine-tuning each contribute meaningfully to the improvement.

**Weaknesses:**

1. The paper assumes TCNS generates realistic false triples, but it provides no quantitative or qualitative validation (e.g., semantic plausibility, frequency bias). If the negatives are too easy or too artificial, the fine-tuned model may learn dataset artifacts rather than genuine fact discrimination.

2. While GRPO is used for paragraph generation, the paper lacks details on reward design, optimization stability, and variance control. Since RLHF-style optimization can be unstable, this omission weakens reproducibility.

3. The experiments mainly involve fact-centric QA and sentence verification tasks, but not open-ended generation or reasoning benchmarks. Moreover, comparisons with existing factual enhancement methods—like retrieval-augmented fine-tuning, factual consistency training, or knowledge editing—are missing.

**Questions:**

How does TCNS handle multi-relation entities where a type-constrained substitution might still form a true triple (i.e., false negatives)?

Did you observe any training instability or reward hacking during GRPO paragraph generation?

---

### Official Review · Reviewer_w2CD · 2025-10-30

**Soundness:** 2
**Presentation:** 2
**Contribution:** 2
**Rating:** 4
**Confidence:** 4

**Summary:**

The paper examines the fact-checking capabilities of large language models (LLMs) by designing three scenarios: assessing whether the model can identify the correctness of original triples, constructing two paragraphs of different difficulty levels based on these triples to evaluate fact-checking ability in long texts, and proposing the use of SFT to enhance the model’s capacity to recognize factual knowledge.

**Strengths:**

1. The paper addresses an important problem — how to enhance the fact-checking capability of LLMs.

2. It leverages existing triple-based datasets to construct more challenging long-text samples that contain subtle factual errors. These samples serve two purposes: evaluating the model’s fact-checking ability and providing training data to further improve that ability.

3. The paper validates the fact-checking performance of LLMs under multiple experimental settings. The results demonstrate that the constructed data are indeed more challenging, and that training on them leads to improved model performance.

**Weaknesses:**

1. The title Enhancing LLM Factuality may mislead readers into thinking the paper focuses on improving the factuality of model responses and reducing hallucinations. However, the actual work is about fact-checking, not response factuality enhancement.

2. The novelty of the paper is limited. The main contribution appears to be introducing GRPO to embed incorrect triple-based facts into the context, thereby generating more challenging detection data. These harder examples are then used for SFT training. However, the consistency between this constructed data and the texts encountered in real-world fact-checking scenarios remains questionable.

3. The writing quality could be improved. The table naming is confusing — entries in the same row are not on the same conceptual level. For example, Topline, Zero-Shot Generated Paragraphs, and GRPO Generated Paragraphs do not correspond: the latter two describe methods of text generation, while “Topline” does not describe a text type. It would be clearer to use something like “Triples” instead. Similarly, Simple Prompt does not match the other items in its row — the others appear to be model names used for text generation (though this is not clearly introduced), while Simple Prompt seems to describe the prompt used by the classifier (again, inferred). This inconsistency makes the table difficult to interpret.

4. There is not introduction about metrics which specifies how scores were calculated. Combined with the confusing table structure, the results are hard to understand and took time to parse. The numbers in the table also lack visual emphasis (e.g., bolding key results), leaving readers unsure where to focus.

5. The figures could also be improved. Their current presentation is not very clear, and understanding them requires referring to specific experimental settings. For example, in Figure 2 (top), a concrete example should illustrate how the swapping process works. In the bottom part, the process of transforming triples into paragraphs should be explicitly described.

6. The paper lacks baseline comparisons with existing fact-checking models or benchmarks.

**Questions:**

See Weaknesses.

---

### Official Review · Reviewer_1PUt · 2025-11-06

**Soundness:** 2
**Presentation:** 2
**Contribution:** 3
**Rating:** 4
**Confidence:** 3

**Summary:**

The paper proposes a data-centric recipe to boost LLM factuality when truth lives in structured sources: generate type-constrained negative samples (plausible-but-fake triples from a knowledge graph), embed them into long, natural paragraphs, then fine-tune models to spot and correct the fakes.
Paragraph generation is trained with a reward that makes the paragraph fool a “student” prompt (no KB fact given) but not an “oracle” prompt (given the source fact), ensuring the fake is present and hard to spot.
Using WebNLG and REBEL, the authors show that GRPO-generated paragraphs are harder and more realistic than simple zero-shot generations, and that training on these challenging cases measurably improves fake-fact detection and correct-fact recognition across LLMs.

**Strengths:**

- The problem of detecting information that is incosistent with a KB can be an important task and could have several potential real life use-cases.
- The proposed recipe, seems to show an improvement both on being able to generate more convincing passages with fake info for models to identify. And using these negatively generated samples shows improvement on teaching the LM to detect fake facts.
- The proposed GRPO self play style algorithm is interesting

**Weaknesses:**

- I found the results section particularly hard to follow. This might be improved with better describing the hypothesis the authors wish to validate with the experiments and how the results validate them.
- If the goal is to show that models detect if text is consistent with a KB, perhaps general purpose KBs may not be the best way to test this as LLMs contain a lot of world info and are probably more likely to suffer on tail facts. Perhaps experiments on more specialized KBs could be more informative (eg, biomedical KBs or some technical ontologies) where the owing to its niche the LLMs initial information might be less likely to bias it.
- Overall the paper is interesting but I believe the experimental section is not very convincing and the paper may benefit from a round of revision. I am happy to reconsider my scores during the rebuttal phase.

**Questions:**

In Tab 1, row 1, does a low topline score indicate that the model does not really know the true facts? If this is the case, wouldn't a good baseline be some continual training with passages with positive facts to see if that improves performance more than negative sampling?

---

### Note · Authors · 2025-11-25

**Comment:**

Thank you to all of the reviewers for the helpful comments and suggestions. We will take care to incorporate these insights into our future work. We appreciate your time and consideration.

**Withdrawal Confirmation:**

I have read and agree with the venue's withdrawal policy on behalf of myself and my co-authors.